# Thermophysical Properties of NH_3_/IL+ Carbon Nanomaterial Solutions

**DOI:** 10.3390/nano11102612

**Published:** 2021-10-04

**Authors:** Gabriela Huminic, Angel Huminic

**Affiliations:** Mechanical Engineering Department, Transilvania University of Brasov, 29, Bulevardul Eroilor, 500036 Brasov, Romania; angel.h@unitbv.ro

**Keywords:** ammonia, ionic liquid, carbon nanomaterials

## Abstract

This study proposes the use of new working fluids, refrigerant/IL+ carbon nanomaterials (CNMs), in absorption systems as an alternative to conventional working fluids. In this regard, the thermophysical properties of ammonia and carbon nanomaterials (graphene and single-wall carbon nanotubes) dispersed into [B_MIM_]BF_4_ ionic liquid are theoretically investigated. The thermophysical properties of NH_3_/IL+ CNMs solutions are computed for weight fractions of NH_3_ in the range of 0.018–0.404 and temperatures between 293 and 388 K. In addition, two weight fractions of CNMs are considered: 0.005 and 0.01, respectively. Our results indicate that by adding a small amount of nanomaterial to the ionic liquid, the solution’s thermal conductivity is enhanced, while its viscosity and specific heat are reduced. Correlations of the thermal conductivity, viscosity, specific heat, and density of the NH_3_/IL+ CNMs solutions are proposed.

## 1. Introduction

Ionic liquids (ILs) are considered a novel type of green working fluid used in various fields, such as absorption refrigeration, solar applications, chemistry (gas capture, storage), and electrochemistry (batteries, sensors). In recent years, ionic liquids have been considered a promising alternative to the conventional working fluids (NH_3_/H_2_O and H_2_O/LiBr) used as absorbents in absorption refrigeration systems due to their good thermal stability, high absorption capacity, and very low vapor pressure [1,2,3].

The thermophysical properties of solutions containing ionic liquids may influence their application in absorption refrigeration systems. Refrigerant/ionic liquid solutions are studied in the literature by way of thermodynamic models, mainly equations of state or activity models [4,5,6,7,8,9,10,11,12,13,14,15]. In addition, two studies on the use of ionic liquids as absorbents have been published by Shiflett and Yokozeki [16,17].

In one paper, Yokozeki and Shiflett [7] carried out a study on the performance of an absorption refrigeration system using NH_3_ as the refrigerant and various ionic liquids as absorbents ([Bmim][PF_6_], [Hmim]Cl, [Bmim][BF_4_], [Emim][SCN], [DMEA][Ac]). The results indicated that the COPs of all the studied solutions were lower than those of the NH_3_/H_2_O solution.

The thermophysical properties (vapor pressures and heat capacities) of the H_2_O + ([Dmim]dmp) system were investigated by Dong et al. [8]. The results revealed that the coefficient of the performance of the H_2_O + [Dmim]dmp system is close to that of the conventional working pair H_2_O + LiBr system.

Kim et al. [9] theoretically investigated the thermodynamic performance of a miniature absorption system using various refrigerant mixtures (R125, R152a, R32, R134a, R143a) /ILs ([Emim][Tf_2_N], [Emim][BF_4_], [Bmim][BF_4_], [Bmim][PF_6_], [Hmim][Tf_2_N], [Hmim][BF_4_], [Hmim][PF_6_]) as the working fluids. They found that refrigerant/IL solutions were promising materials for absorption refrigeration systems that utilize low-grade waste heat, such as those of electronic systems.

Kim and Kohl [10] carried out an analysis of the performance of R134a/[Bmim][PF_6_] and R134a/[Hmim][Tf_2_N] using the Redlich–Kwong equation of state and a two-phase pressure drop model. They noticed that R134a/[Hmim][Tf_2_N] exhibited a higher system efficiency compared to R134a [Hmim][PF_6_], except in the case where the solubility difference between the absorber and desorber converged to zero.

In another paper, Kim and Kohl [11] investigated the cooling capability of the R134/[Bmim][PF_6_] used in an absorption refrigeration system. They compared the performance of R134/[Bmim][PF_6_] with R134a/[Bmim][PF_6_] and found that R134/[Bmim][PF_6_] had a 1.9 times higher cooling capability than R134a/[Bmim][PF_6_] at a desorber temperature as low as 63 °C. In addition, R134/[Bmim][PF_6_] had a coefficient of performance up to three times higher than that of R134a/[Bmim][PF_6_]. Chen and Bay [18] investigated the thermal performance of an absorption refrigeration system using [Emim]Cu_2_Cl_5_/NH_3_ and found that the thermal performance of the [Emim]Cu_2_Cl_5_/NH_3_ solution was better than that of a NH_3_/H_2_O solution, but slightly lower than that of a LiBr/H_2_O solution. In another study, Chen et al. [19] numerically investigated the thermodynamic performance of an absorption system using [Bmim]Zn_2_Cl_5_/NH_3_. The results revealed that that the [Bmim]Zn_2_Cl_5_/NH_3_ absorption system exhibited higher thermal performance compared to a NaSCN/NH_3_ absorption system.

The thermodynamic performance of an absorption chiller using [Emim][dmp]/H_2_O was simulated by Zhang and Hu [20]. Their results showed that the coefficient of performance was lower than that of a H_2_O/LiBr solution, concluding that [Emim][dmp] may be a good absorbent for refrigeration systems. Martin et al. [21] carried out a study on the use of ILs with supercritical CO_2_ using a group contribution equation of state and found that the coefficient of performance was lower compared to a conventional NH_3_/H_2_O system.

Table 1 presents the values of the coefficients of performance for absorption refrigeration systems using ammonia/ionic liquids as working fluids.

Investigations into the application of ammonia/ionic liquids as working fluids in absorption refrigeration systems are limited in the open literature. Moreover, studies on absorption refrigeration systems using ammonia/ionic liquid+nanomaterials as working fluids are not reported in the literature. In order to improve the performance of absorption systems, new working fluids are herein proposed. The thermophysical properties of working fluids are the main data in this evaluation of the performance of absorption refrigeration systems. In this regard, the thermophysical properties of ammonia with graphene (GE) and single-wall carbon nanotubes (SWCNTs), respectively, dispersed into [Bmim]BF_4_ ionic liquid, are analyzed and discussed. Correlations for the studied properties, required for the modeling and simulation of the performance of various absorption refrigeration systems, are also proposed.

## 2. Thermophysical Properties of the Solutions

The thermophysical properties of the working fluids used in absorption refrigeration systems must be determined as an essential step in the evaluation of the thermodynamic performance of these fluids. In this study, ammonia and two types of carbon nanomaterials (CNMs—graphene (GE) and single-wall carbon nanotubes (SWCNTs)) with two weight fractions (0.005 and 0.01), dispersed into a [B_mim_]BF_4_ ionic liquid, will be analyzed and discussed. The thermophysical properties of ammonia and CMNs/[B_mim_]BF_4_ were taken from the NIST database [24] and Fang et al. [25], respectively.

Since there are no data on the thermo-properties of IL+CNMs mixed with NH_3_ solutions, the properties (thermal conductivity, specific heat, and density) were calculated using a general equation, based on the weighted average of the properties of both components of the mixture [26,27]:(1)$$M_{sol}=w_{NH_{3}}M_{NH_{3}}+w_{IL}M_{IL+CNMs}$$
in which the mass fraction of NH_3_ is calculated as:(2)$$w_{NH_{3}}=\frac{x_{A}M_{A}}{x_{A}M_{A}+x_{B}M_{B}}\;$$

The solution dynamic viscosity is calculated as:(3)$$\ln\mu_{sol}=w_{NH_{3}}\ln\mu_{NH_{3}}+\left(1-w_{NH_{3}}\right)\ln\mu_{IL+CNMs}$$

## 3. Results and Discussions

In this study, the thermo-properties of the NH_3_/[B_mim_]BF_4_ and NH_3_/[B_mim_]BF_4_+ CNMs solutions were evaluated for the mass fractions of NH_3_ in a range of 0.018–0.404 and at temperatures from 293 K to 388 K. Two types of carbon nanomaterials with two weight fractions (0.005 and 0.01), dispersed into an ionic liquid, were considered: graphene (GE) and single-wall carbon nanotubes (SWCNTs). No data for the thermal properties of the [B_mim_]BF_4_ +CNMs mixed with NH_3_ solutions have been reported in the literature.

### 3.1. Thermal Conductivity

Figure 1a–e shows the variation of the thermal conductivity of NH_3_/[B_mim_]BF_4_, NH_3_/[B_mim_]BF_4_ + GE and NH_3_/[B_mim_]BF_4_ + SWCNTs with the temperature at various NH_3_ fractions. With increasing temperatures can be seen that the thermal conductivities of all solutions have an upward trend up to w_NH_3_ = 0.048, then with increasing NH_3_ fractions (≥0.102), the thermal conductivities decrease with increasing temperatures, but increasing with increasing NH_3_ fractions. The addition of carbon nanomaterials to the ionic liquid leads to an enhancement in the solution’s thermal conductivity compared to the base solution. The enhancements in the thermal conductivity of the studied solutions—calculated as $\left[(k_{NH_{3}/Il+CNMs}-k_{NH_{3}/Il})/k_{NH_{3}/Il}\right]\times 100$—at minimum and maximum NH_3_ fractions—$w_{NH3}=0.018$ and $w_{NH3}=0.404$, respectively—are shown in Table 2:

It can be seen from Table 2 that the maximum enhancement in thermal conductivity was achieved by$\;NH_{3}/\left[B_{mim}\right]BF_{4}+0.01\;GE$, while the minimum enhancement in thermal conductivity was recorded for $NH_{3}/\left[B_{mim}\right]BF_{4}+0.005\;SWCNTs$ for both NH_3_ fractions. In addition, a descending trend in the thermal conductivity of solutions with higher NH_3_ fractions may be seen. These results may be explained by the thermal conductivities of carbon nanomaterials dispersed into the ionic liquid. Graphene (GE) exhibits a thermal conductivity of ~4000 W/mK [28,29,30], while the thermal conductivity of SWCNTs is usually reported to be in the range of 2000–6000 W/mK at a standard temperature (25 °C) [31]. Yu et al. [32] measured the thermal conductivity of SWCNTs using a chemical vapor deposition method and found a value higher than 2000 W/mK. The experimental results related to the thermal conductivity of ionic liquids revealed the increase in thermal conductivity achieved by adding nanoparticles into the ionic liquid and the minor influence of temperature on several ionic liquids containing nanoparticles. The main arguments for these trends are thermal boundary resistance, layering phenomena, and clustering [33].

The thermal conductivity values are correlated by means of a linear equation as a function of temperature:(4)k=aT+b

The coefficient values a, b, and $R^{2}$ are given in Table 3.

### 3.2. Dynamic Viscosity

Figure 2a–e depict the variation in the viscosity of the solutions, with various NH_3_ fractions, at rising temperatures. As shown in Figure 2, the viscosities of the solutions decrease exponentially with higher temperatures. By adding the carbon nanomaterials into the ionic liquid, a reduction in viscosity may be seen compared to the base solution. Higher fractions of carbon nanomaterials lead to an increase in the viscosity of the studied solutions, but these viscosity values do not exceed those of the base solution. With higher NH_3_ fractions, a decrease in viscosity may also be seen. The diminution in viscosity is more obvious at the lower mass fractions of NH_3_ in the solutions. The viscosities of the NH_3_/IL+CNMs solutions are lower than that of the base solution, indicating that these solutions are suitable for NH_3_ absorption. The reduction in the viscosity of the studied solutions—calculated as $\left[(\mu_{NH_{3}/Il}-\mu_{NH_{3}/Il+CNMs})/(\mu_{NH_{3}/Il}\right]\times 100$—at minimum and maximum NH_3_ fractions—$w_{NH3}=0.018$ and $w_{NH3}=0.404$, respectively—is shown in Table 4:

From Table 4, it may be seen that at a temperature of 293 K the maximum reduction in viscosity is achieved by$\;NH_{3}/\left[B_{mim}\right]BF_{4}+0.01\;GE$, while the minimum is seen in the case of $NH_{3}/\left[B_{mim}\right]BF_{4}+0.01\;SWCNTs$ for both NH_3_ fractions. With increasing temperature, $NH_{3}/\left[B_{mim}\right]BF_{4}+0.01\;GE$ and $NH_{3}/\left[B_{mim}\right]BF_{4}+0.005\;SWCNTs$ show similar values of viscosity reduction. In addition, a descending trend in viscosity may be seen at higher NH_3_ fractions in the solutions, with the viscosity values of the NH_3_/IL+CNMs being similar to the values of the base solution.

The data related to the dynamic viscosity of ionic liquids are still contradictory. Most experimental studies indicate an increase in viscosity with the addition of nanoparticles to the ionic liquid, while on the other hand there are studies that have found a decrease in viscosity. The reduction in viscosity can be explained by the interaction between the molecules of the ionic liquid and the nanoparticles, as well as by the lubricating properties of the nanoparticles.

The dynamic viscosity values are correlated by means of an exponential equation as a function of temperature:(5)$$\mu =a\times e^{b\times T}$$

The coefficient values a, b, and $R^{2}$ are given in Table 5.

### 3.3. Specific Heat

Figure 3a–e illustrate the variation in the specific heat of the solutions, with various NH_3_ fractions, at rising temperatures. As shown in Figure 3, the specific heat of the solutions increases with both higher temperatures and higher fractions of NH_3_. In addition, by adding nanoparticles to the ionic liquid, a reduction in specific heat may be seen compared to the base solution. Higher CNMs fractions led to a decrease in the specific heat of all the solutions. The presented results are according to an equation proposed by Raud et al. [34], which indicates the increase in a solution’s specific heat with rising temperatures, and also the reduction in specific heat by the addition of nanomaterials into the base solution.

The reduction in the specific heat of the studied solutions—calculated as $\left[(c_{p,NH_{3}/Il}-c_{p,NH_{3}/Il+CNMs})/(c_{p,NH_{3}/Il}\right]\times 100$—at minimum and maximum NH_3_ fractions—$w_{NH3}=0.018$ and $w_{NH3}=0.404$, respectively—is shown in Table 6:

The maximum reduction in specific heat was achieved by both solutions with a 0.01 fraction of nanomaterials, $\;NH_{3}/\left[B_{mim}\right]BF_{4}+0.01\;GE$ and $NH_{3}/\left[B_{mim}\right]BF_{4}+0.01\;SWCNTs)$, while the minimum can be seen in the case of $NH_{3}/\left[B_{mim}\right]BF_{4}+0.005\;GE$, for both NH_3_ fractions, at a temperature of 293 K. With higher temperatures, the maximum reduction was recorded for $NH_{3}/\left[B_{mim}\right]BF_{4}+0.005\;SWCNTs$. With higher fractions of NH_3_, the reduction in specific heat was significant.

The data available in the open literature related to the specific heat of ionic liquids are, as in the case of viscosity, contradictory. The main reasons for this are the interaction between the molecules of the nanomaterials and the ionic liquid and the chemical structure of the ionic liquid.

The specific heat values are correlated by means of a linear equation as a function of temperature:(6)$$c_{p}=aT+b$$

In Table 7, the coefficient values a, b, and $R^{2}$ are given:

### 3.4. Density

The densities of the solutions, with various NH_3_ fractions and at rising temperatures, are illustrated in Figure 4a–e. As can be seen, the density decreases with both higher temperature and higher NH_3_ fractions. The addition of carbon nanomaterials into the ionic liquid increases the solution’s density compared to the base solution. In addition, higher CNMs fractions lead to increased density for all solutions. Most experimental studies regarding the density of ionic liquids report an increase in density with the addition of nanoparticles and a decrease with higher temperatures. The presented results show the same trend as the experimental results obtained by other studies [35].

The enhancement in the density of the studied solutions—calculated as $\left[(\rho_{NH_{3}/Il+CNMs}-\rho_{NH_{3}/Il})/(\rho_{NH_{3}/Il}\right]\times 100$—at minimum and maximum NH_3_ fractions—$w_{NH3}=0.018$ and $w_{NH3}=0.404$, respectively—is shown in Table 8:

At a temperature of 293 K, the maximum enhancement in density was achieved by $NH_{3}/\left[B_{mim}\right]BF_{4}+0.01\;GE$, while the minimum can be seen in the case of $NH_{3}/\left[B_{mim}\right]BF_{4}+0.005\;GE$ for both NH_3_ fractions. At higher temperatures, the maximum enhancement was recorded for $NH_{3}/\left[B_{mim}\right]BF_{4}+0.005\;SWCNTs$.

The density values are correlated by means of a linear equation as a function of temperature:(7)ρ=aT+b

The coefficient values a, b, and $R^{2}$ are given in Table 9.

## 4. Conclusions

In this study, the thermophysical properties of ammonia and carbon nanomaterials (CNMs), dispersed into [B_mim_]BF_4_ ionic liquid, were analyzed and discussed. The results showed that the thermal conductivity of the solutions decreases with higher NH_3_ fractions. By adding carbon nanomaterials into the ionic liquid, an enhancement in the solution’s thermal conductivity may be seen compared to the base solution, with the maximum enhancement in thermal conductivity having been achieved by$\;NH_{3}/\left[B_{mim}\right]BF_{4}+0.01\;GE$. The viscosities of the NH_3_/IL+CMNs solutions were lower than that of the base solution, indicating that these solutions are suitable for NH_3_ absorption. In this case, the maximum reduction in viscosity was recorded for$\;NH_{3}/\left[B_{mim}\right]BF_{4}+0.01\;GE$. In addition, by adding CMNs to the ionic liquid, a reduction in the specific heat of the solutions may be seen compared to the base solution. At a temperature of 293 K, the maximum reduction in specific heat was achieved by the solutions with a 0.01 fraction of nanomaterials ($NH_{3}/\left[B_{mim}\right]BF_{4}+0.01\;GE$ and $NH_{3}/\left[B_{mim}\right]BF_{4}+0.01\;SWCNTs)$. Moreover, the addition of CMNs to the ionic liquid led to an increase in the solution’s density. At a temperature of 293 K, the maximum enhancement in density was achieved by$\;NH_{3}/\left[B_{mim}\right]BF_{4}+0.01\;GE$. Finally, correlations for all studied properties were proposed.

The results of this study may contribute to the consolidation of the property database of NH_3_/IL+NMs for applications in absorption refrigeration. Further investigations concerning the thermophysical characteristics of ammonia with other types of ionic liquids are needed. In addition, for the practical implementation of NH_3_/ILs+CNMs in absorption refrigeration systems, experimental studies to support the reported theoretical results are needed.

## Figures and Tables

**Figure 1 nanomaterials-11-02612-f001:**
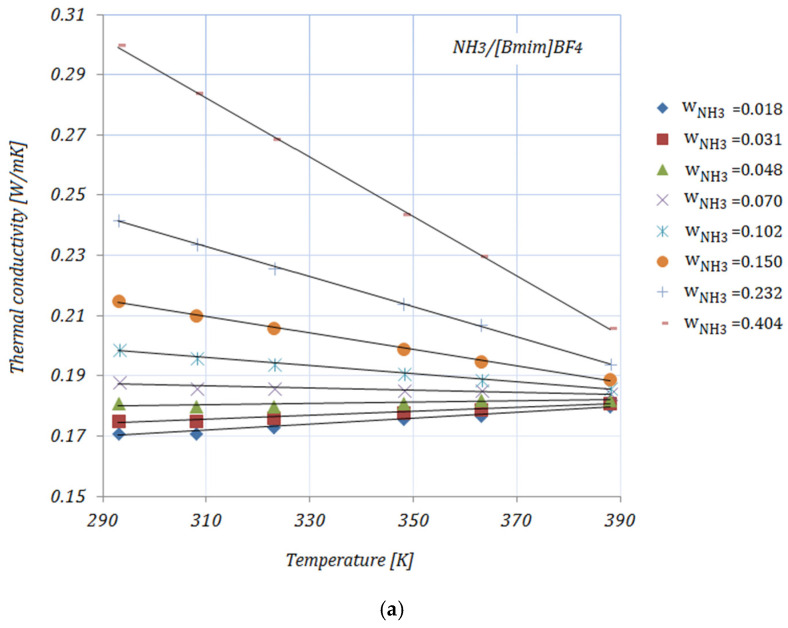
Solution thermal conductivity.

**Figure 2 nanomaterials-11-02612-f002:**
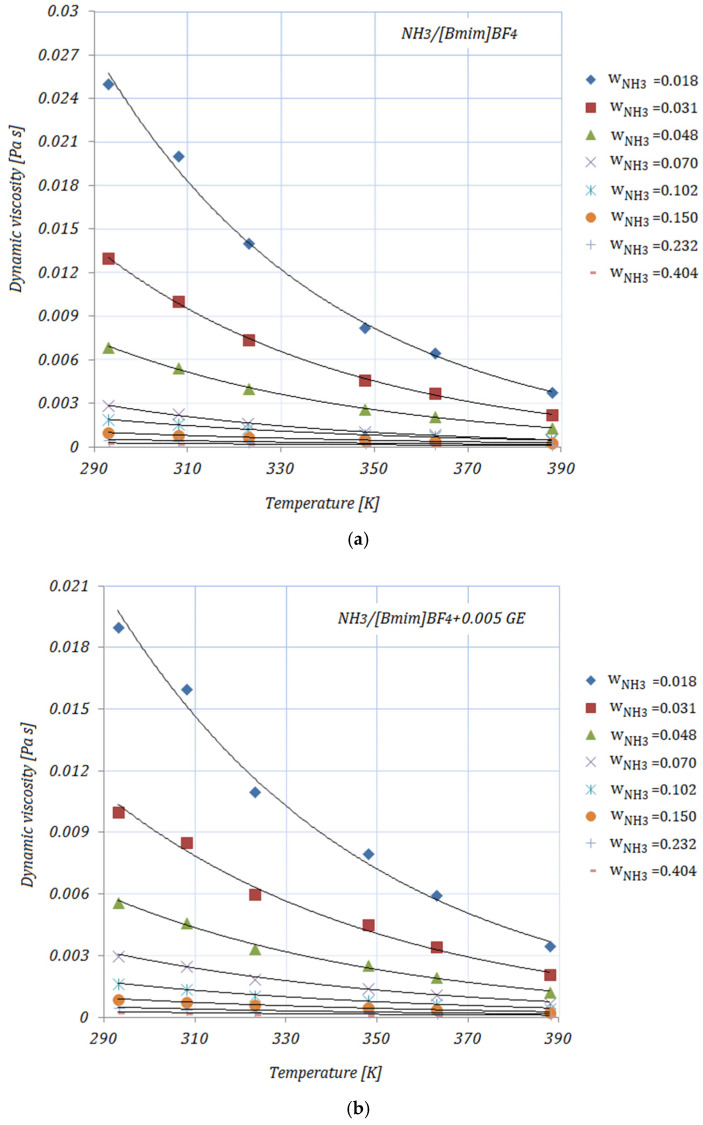
Solution dynamic viscosity.

**Figure 3 nanomaterials-11-02612-f003:**
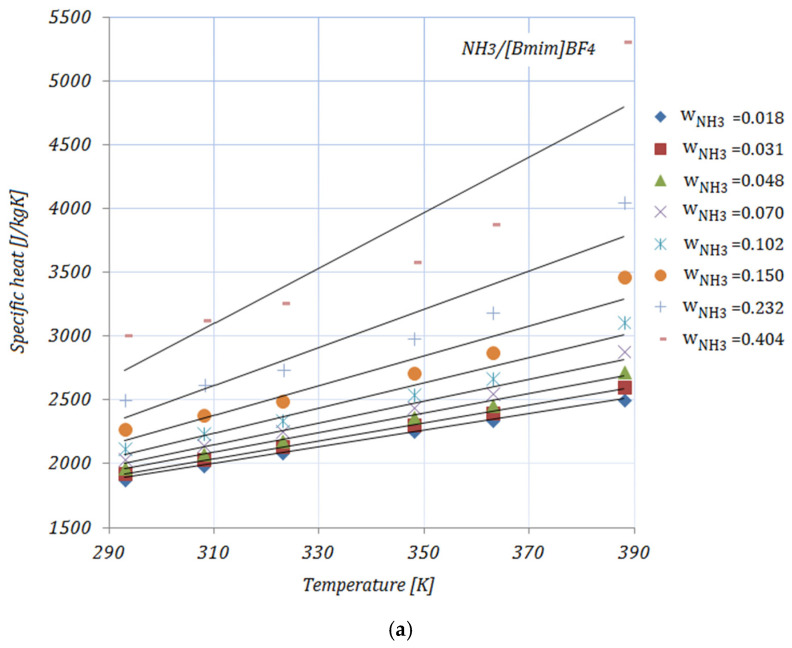
Solution specific heat.

**Figure 4 nanomaterials-11-02612-f004:**
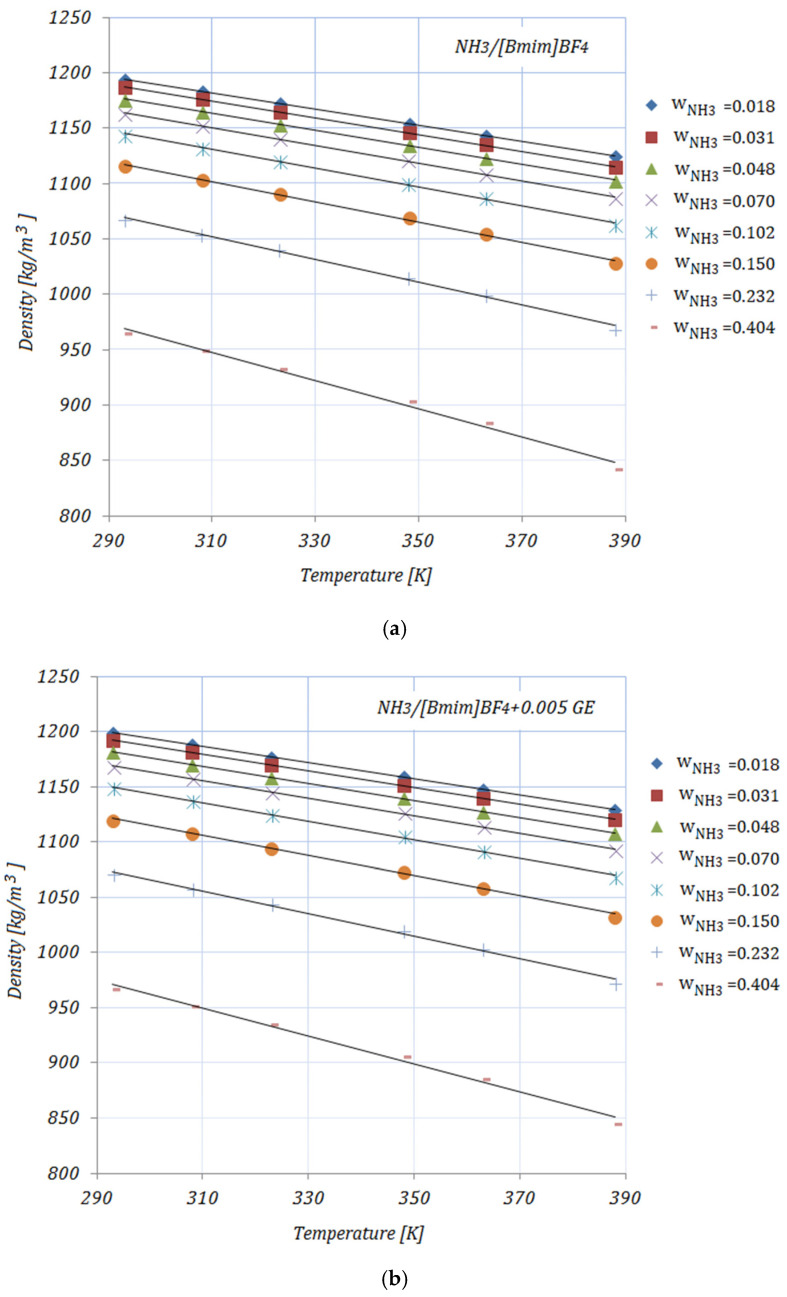
Solution density.

**Table 1 nanomaterials-11-02612-t001:** The coefficients of performance of absorption refrigeration systems using various NH_3_/Ionic liquids.

NH_3_/Ionic Liquid	Ariyadi [22]	Yokozeki, M.B. Shiflett [6,7]	Ferro et al. [23]	Chen and Bay [18]
	Coefficient of Performance (COP)			
NH_3_/[Bmim][BF_4_]	0.715	0.557		
NH_3_/[Bmim][PF_6_]	0.588	0.575		
NH_3_/[Emim][Tf_2_N]	0.657	0.589		
NH_3_/[Emim][EtOSO_3_]	0.612	0.485	0.540	
NH_3_/[Emim][SCN]	0.648	0.557	0.592	
NH_3_/[DMEA][Ac]		0.612		
NH_3_/[Emim][Ac]		0.573	0.644	
NH_3_/[Emim]Cu_2_Cl_5_				0.781
NH_3_/[Choline][NTf_2_]			0.668	
NH_3_/Water		0.646		

**Table 2 nanomaterials-11-02612-t002:** Enhancements in thermal conductivity.

*Temp. [K]*	$NH_{3}/\left[B_{mim}\right]BF_{4}+0.005\;GE$		$NH_{3}/\left[B_{mim}\right]BF_{4}+0.01\;GE$		$NH_{3}/\left[B_{mim}\right]BF_{4}+0.005\;SWCNTs$		$NH_{3}/\left[B_{mim}\right]BF_{4}+0.01\;SWCNTs$	
	$w_{NH3}=0.018$	$w_{NH3}=0.404$	$w_{NH3}=0.018$	$w_{NH3}=0.404$	$w_{NH3}=0.018$	$w_{NH3}=0.404$	$w_{NH3}=0.018$	$w_{NH3}=0.404$
	*Enhancement [%]*							
293	12.28	4.33	14.61	5.33	5.26	2.00	13.45	4.66
388	14.44	7.28	15.55	7.76	5.55	2.91	14.45	7.30

**Table 3 nanomaterials-11-02612-t003:** Coefficient values a, b, and $R^{2}$ obtained by fitting Equation (4).

** *Solution* **	$NH_{3}/\left[B_{mim}\right]BF_{4}$			$NH_{3}/\left[B_{mim}\right]BF_{4}+0.005\;GE$			$NH_{3}/\left[B_{mim}\right]BF_{4}+0.01\;GE$		
** *Coefficients* **	** *a* **	** *b* **	$R^{2}$	** *a* **	** *b* **	$R^{2}$	** *a* **	** *b* **	$R^{2}$
$w_{{\mathrm{NH}}_{3}}=0.018$	0.0001	0.1409	0.979	0.0002	0.1468	0.994	0.0001	0.164	0.674
$w_{{\mathrm{NH}}_{3}}=0.031$	$7\times 10^{-5}$	0.1548	0.974	0.0001	0.1618	0.975	$7\times 10^{-5}$	0.1811	0.460
$w_{{\mathrm{NH}}_{3}}=0.048$	$2\times 10^{-5}$	0.1746	0.565	$6\times 10^{-5}$	0.1833	0.901	$3\times 10^{-5}$	0.1972	0.158
$w_{{\mathrm{NH}}_{3}}=0.070$	$4\times 10^{-5}$	0.1977	0.864	$3\times 10^{-6}$	0.2066	0.032	$-3\times 10^{-5}$	0.2218	0.175
$w_{{\mathrm{NH}}_{3}}=0.102$	−0.0001	0.2374	0.993	$-9\times 10^{-5}$	0.2424	0.957	−0.0001	0.2592	0.794
$w_{{\mathrm{NH}}_{3}}=0.150$	−0.0003	0.2943	0.998	−0.0002	0.3008	0.987	−0.0003	0.3148	0.946
$w_{{\mathrm{NH}}_{3}}=0.232$	−0.0005	0.3883	0.999	−0.0005	0.3948	0.998	−0.0005	0.4065	0.986
$w_{{\mathrm{NH}}_{3}}=0.404$	−0.001	0.5883	0.999	−0.001	0.5939	0.999	−0.001	0.6059	0.998
** *Solution* **	$NH_{3}/\left[B_{mim}\right]BF_{4}+0.005\;SWCNTs$				$NH_{3}/\left[B_{mim}\right]BF_{4}+0.01\;SWCNTs$				
** *Coefficients* **	** *a* **	** *b* **		$R^{2}$	** *a* **		** *b* **	$R^{2}$	
$w_{{\mathrm{NH}}_{3}}=0.018$	0.0001	0.1429		0.897	0.0001		0.1545	0.993	
$w_{{\mathrm{NH}}_{3}}=0.031$	$9\times 10^{-5}$	0.1567		0.875	0.0001		0.167	0.962	
$w_{{\mathrm{NH}}_{3}}=0.048$	$4\times 10^{-5}$	0.1778		0.461	$5\times 10^{-5}$		0.1885	0.927	
$w_{{\mathrm{NH}}_{3}}=0.070$	$-2\times 10^{-5}$	0.2024		0.277	$-2\times 10^{-5}$		0.2143	0.536	
$w_{{\mathrm{NH}}_{3}}=0.102$	−0.0001	0.2383		0.890	−0.0001		0.2501	0.966	
$w_{{\mathrm{NH}}_{3}}=0.150$	−0.0002	0.2944		0.977	−0.0002		0.3061	0.996	
$w_{{\mathrm{NH}}_{3}}=0.232$	−0.0005	0.3919		0.994	−0.0005		0.397	0.998	
$w_{{\mathrm{NH}}_{3}}=0.404$	−0.001	0.59		0.998	−0.001x		0.5981	0.999	

**Table 4 nanomaterials-11-02612-t004:** Reduction in dynamic viscosity.

*Temp. [K]*	$NH_{3}/\left[B_{mim}\right]BF_{4}+0.005\;GE$		$NH_{3}/\left[B_{mim}\right]BF_{4}+0.01\;GE$		$NH_{3}/\left[B_{mim}\right]BF_{4}+0.005\;SWCNTs$		$NH_{3}/\left[B_{mim}\right]BF_{4}+0.01\;SWCNTs$	
	$w_{NH3}=0.018$	$w_{NH3}=0.404$	$w_{NH3}=0.018$	$w_{NH3}=0.404$	$w_{NH3}=0.018$	$w_{NH3}=0.404$	$w_{NH3}=0.018$	$w_{NH3}=0.404$
	*Reduction [%]*							
293	24.0	3.35	36.0	5.65	28.0	4.44	8.00	1.24
388	7.31	0.95	14.64	1.99	14.86	1.98	0.026	0.36

**Table 5 nanomaterials-11-02612-t005:** Coefficient values a, b, and $R^{2}$ obtained by fitting Equation (5).

** *Solution* **	$NH_{3}/\left[B_{mim}\right]BF_{4}$			$NH_{3}/\left[B_{mim}\right]BF_{4}+0.005\;GE$			$NH_{3}/\left[B_{mim}\right]BF_{4}+0.01\;GE$		
** *Coefficients* **	** *a* **	** *b* **	$R^{2}\;$	** *a* **	** *b* **	$R^{2}\;$	** *a* **	** *b* **	$R^{2}\;$
$w_{{\mathrm{NH}}_{3}}=0.018$	9.4532	−0.020	0.998	3.5308	−0.018	0.992	2.3435	−0.017	0.998
$w_{{\mathrm{NH}}_{3}}=0.031$	3.0038	−0.019	0.999	1.2734	−0.016	0.991	1.0015	−0.016	0.998
$w_{{\mathrm{NH}}_{3}}=0.048$	1.1949	−0.018	0.999	0.5719	−0.016	0.993	0.4298	−0.015	0.998
$w_{{\mathrm{NH}}_{3}}=0.070$	0.6608	−0.019	0.999	0.2337	−0.015	0.994	0.1844	−0.014	0.998
$w_{{\mathrm{NH}}_{3}}=0.102$	0.1567	−0.015	0.999	0.0957	−0.014	0.995	0.0791	−0.013	0.998
$w_{{\mathrm{NH}}_{3}}=0.150$	0.0566	−0.014	0.999	0.0391	−0.013	0.996	0.0791	−0.013	0.998
$w_{{\mathrm{NH}}_{3}}=0.232$	0.0205	−0.013	0.999	0.016	−0.012	0.997	0.0145	−0.012	0.998
$w_{{\mathrm{NH}}_{3}}=0.404$	0.0074	−0.011	0.998	0.0065	−0.011	0.997	0.0062	−0.011	0.998
** *Solution* **	$NH_{3}/\left[B_{mim}\right]BF_{4}+0.005\;SWCNTs$				$NH_{3}/\left[B_{mim}\right]BF_{4}+0.01\;SWCNTs$				
** *Coefficients* **	** *a* **	** *b* **		$R^{2}$	** *a* **		** *b* **	$R^{2}$	
$w_{{\mathrm{NH}}_{3}}=0.018$	3.207	−0.018		0.993	6.8882		−0.019	0.997	
$w_{{\mathrm{NH}}_{3}}=0.031$	1.2694	−0.017		0.994	2.5225		−0.018	0.997	
$w_{{\mathrm{NH}}_{3}}=0.048$	0.5266	−0.016		0.994	0.9613		−0.017	0.998	
$w_{{\mathrm{NH}}_{3}}=0.070$	0.2184	−0.015		0.995	0.3606		−0.016	0.999	
$w_{{\mathrm{NH}}_{3}}=0.102$	0.0905	−0.014		0.995	0.1352		−0.015	0.999	
$w_{{\mathrm{NH}}_{3}}=0.150$	0.0375	−0.013		0.996	0.0507		−0.014	0.999	
$w_{{\mathrm{NH}}_{3}}=0.232$	0.0156	−0.012		0.997	0.0190		−0.012	0.999	
$w_{{\mathrm{NH}}_{3}}=0.404$	0.0065	−0.011		0.997	0.0071		−0.011	0.999	

**Table 6 nanomaterials-11-02612-t006:** Reduction in specific heat.

*Temp. [K]*	$NH_{3}/\left[B_{mim}\right]BF_{4}+0.005\;GE$		$NH_{3}/\left[B_{mim}\right]BF_{4}+0.01\;GE$		$NH_{3}/\left[B_{mim}\right]BF_{4}+0.005\;SWCNTs$		$NH_{3}/\left[B_{mim}\right]BF_{4}+0.01\;SWCNTs$	
	$w_{NH3}=0.018$	$w_{NH3}=0.404$	$w_{NH3}=0.018$	$w_{NH3}=0.404$	$w_{NH3}=0.018$	$w_{NH3}=0.404$	$w_{NH3}=0.018$	$w_{NH3}=0.404$
	*Reduction [%]*							
293	11.26	4.25	14.39	5.35	12.32	4.75	14.40	5.45
388	11.09	3.11	10.89	2.92	12.45	3.65	10.90	3.05

**Table 7 nanomaterials-11-02612-t007:** Coefficient values a, b, and $R^{2}$ obtained by fitting Equation (6).

** *Solution* **	$NH_{3}/\left[B_{mim}\right]BF_{4}$			$NH_{3}/\left[B_{mim}\right]BF_{4}+0.005\;GE$			$NH_{3}/\left[B_{mim}\right]BF_{4}+0.01\;GE$		
** *Coefficients* **	** *a* **	** *b* **	$R^{2}\;$	** *a* **	** *b* **	$R^{2}\;$	** *a* **	** *b* **	$R^{2}\;$
$w_{{\mathrm{NH}}_{3}}=0.018$	6.5091	−14.982	0.999	5.7547	−6.8071	0.998	6.4912	−288.77	0.999
$w_{{\mathrm{NH}}_{3}}=0.031$	7.0161	−134.92	0.998	6.2453	−122.19	0.996	7.0114	−409.34	0.997
$w_{{\mathrm{NH}}_{3}}=0.048$	7.6829	−293.09	0.992	6.9265	−280.88	0.987	7.6434	−555.11	0.989
$w_{{\mathrm{NH}}_{3}}=0.070$	8.5537	−500.19	0.980	7.8132	−488.02	0.972	8.5449	−762.4	0.974
$w_{{\mathrm{NH}}_{3}}=0.102$	9.8166	−800.83	0.958	9.0946	−786.39	0.947	9.7737	−1046	0.953
$w_{{\mathrm{NH}}_{3}}=0.150$	11.706	−1249.5	0.927	11.026	−1237.4	0.914	11.71	−1492.8	0.921
$w_{{\mathrm{NH}}_{3}}=0.232$	14.945	−2020.6	0.884	14.328	−2008.4	0.872	14.943	−2237.7	0.879
$w_{{\mathrm{NH}}_{3}}=0.404$	21.718	−3629.6	0.830	21.283	−3630.8	0.821	21.773	−3812.5	0.826
** *Solution* **	$NH_{3}/\left[B_{mim}\right]BF_{4}+0.005\;SWCNTs$				$NH_{3}/\left[B_{mim}\right]BF_{4}+0.01\;SWCNTs$				
** *Coefficients* **	** *a* **	** *b* **		$R^{2}$	** *a* **		** *b* **	$R^{2}$	
$w_{{\mathrm{NH}}_{3}}=0.018$	5.6603	−8.2981		0.999	6.4453		−282.79	0.998	
$w_{{\mathrm{NH}}_{3}}=0.031$	6.1439	−121.18		0.995	6.9283		−395.16	0.993	
$w_{{\mathrm{NH}}_{3}}=0.048$	6.832	−281.37		0.985	7.560		−549.92	0.982	
$w_{{\mathrm{NH}}_{3}}=0.070$	7.7137	−486.29		0.968	8.4608		−747.21	0.966	
$w_{{\mathrm{NH}}_{3}}=0.102$	9.003	−786.18		0.942	9.7313		−1040.7	0.943	
$w_{{\mathrm{NH}}_{3}}=0.150$	10.946	−1239.4		0.909	11.63		−1478.2	0.912	
$w_{{\mathrm{NH}}_{3}}=0.232$	14.247	−2007.4		0.868	14.869		−2224.7	0.872	
$w_{{\mathrm{NH}}_{3}}=0.404$	21.177	−3619.5		0.819	21.705		−3799.5	0.822	

**Table 8 nanomaterials-11-02612-t008:** Enhancement in density.

*Temp. [K]*	$NH_{3}/\left[B_{mim}\right]BF_{4}+0.005\;GE$		$NH_{3}/\left[B_{mim}\right]BF_{4}+0.01\;GE$		$NH_{3}/\left[B_{mim}\right]BF_{4}+0.005\;SWCNTs$		$NH_{3}/\left[B_{mim}\right]BF_{4}+0.01\;SWCNTs$	
	$w_{NH3}=0.018$	$w_{NH3}=0.404$	$w_{NH3}=0.018$	$w_{NH3}=0.404$	$w_{NH3}=0.018$	$w_{NH3}=0.404$	$w_{NH3}=0.018$	$w_{NH3}=0.404$
	*Enhancement [%]*							
293	0.41	0.26	1.34	25.42	0.083	23.87	0.58	24.49
388	0.35	0.30	1.15	0.81	0.088	0.075	0.62	0.46

**Table 9 nanomaterials-11-02612-t009:** Coefficient values a, b, and $R^{2}$ obtained by fitting Equation (7).

** *Solution* **	$NH_{3}/\left[B_{mim}\right]BF_{4}$			$NH_{3}/\left[B_{mim}\right]BF_{4}+0.005\;GE$			$NH_{3}/\left[B_{mim}\right]BF_{4}+0.01\;GE$		
** *Coefficients* **	** *a* **	** *b* **	$R^{2}\;$	** *a* **	** *b* **	$R^{2}\;$	** *a* **	** *b* **	$R^{2}\;$
$w_{{\mathrm{NH}}_{3}}=0.018$	−0.7283	1407.6	0.999	−0.7341	1414.2	0.999	−0.75	1430.2	0.999
$w_{{\mathrm{NH}}_{3}}=0.031$	−0.7558	1408.8	0.999	−0.7558	1413.8	0.999	−0.77	1428.5	0.998
$w_{{\mathrm{NH}}_{3}}=0.048$	−0.7678	1401.6	0.999	−0.7748	1409.1	0.999	−0.7918	1425.1	0.998
$w_{{\mathrm{NH}}_{3}}=0.070$	−0.7993	1398	0.999	−0.7953	1401.8	0.999	−0.8233	1420.6	0.997
$w_{{\mathrm{NH}}_{3}}=0.102$	−0.844	1392.2	0.998	−0.844	1397.2	0.998	−0.8599	1412.3	0.996
$w_{{\mathrm{NH}}_{3}}=0.150$	−0.9143	1358	0.997	−0.9109	1388.5	0.998	−0.9349	1405	0.995
$w_{{\mathrm{NH}}_{3}}=0.232$	−1.0312	1371.6	0.996	−1.0246	1373.7	0.995	−1.0429	1387.5	0.992
$w_{{\mathrm{NH}}_{3}}=0.404$	−1.2654	1339.4	0.991	−1.265	1341.9	0.991	−1.2802	1352.8	0.989
** *Solution* **	$NH_{3}/\left[B_{mim}\right]BF_{4}+0.005\;SWCNTs$				$NH_{3}/\left[B_{mim}\right]BF_{4}+0.01\;SWCNTs$				
** *Coefficients* **	** *a* **	** *b* **		$R^{2}$	** *a* **		** *b* **	$R^{2}$	
$w_{{\mathrm{NH}}_{3}}=0.018$	−0.7271	1407.7		0.999	−0.7261		1413.6	0.999	
$w_{{\mathrm{NH}}_{3}}=0.031$	−0.7421	1404.7		0.999	−0.7478		1413.3	0.999	
$w_{{\mathrm{NH}}_{3}}=0.048$	−0.7678	1402.6		0.999	−0.7707		1409.9	0.999	
$w_{{\mathrm{NH}}_{3}}=0.070$	−0.801	1399.4		0.999	−0.7953		1403.8	0.999	
$w_{{\mathrm{NH}}_{3}}=0.102$	−0.844	1393.2		0.983	−0.844		1399.2	0.998	
$w_{{\mathrm{NH}}_{3}}=0.150$	−0.9109	1384.5		0.998	−0.9109		1390.5	0.998	
$w_{{\mathrm{NH}}_{3}}=0.232$	−1.0273	1370.9		0.996	−1.0281		1376.3	0.996	
$w_{{\mathrm{NH}}_{3}}=0.404$	−1.2632	1339.2		0.991	−1.2646		1343.1	0.991	

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
