# Peer review of "Thermophysical Properties of NH3/IL+ Carbon Nanomaterial Solutions"

_nanomaterials, 2021, doi:10.3390/nano11102612_

Round 1
Reviewer 1 Report
This manuscript analyzed and discussed the thermo-physical properties of the ammonia with graphene and single-wall carbon nanotubes respectively dispersed into [Bmim]BF4 ionic liquid, systematically. The correlations of the thermal conductivity, viscosity, specific heat, and density of the NH3/IL+ CNMs solutions are proposed. This study may provide a good reference for experimental researches in the ionic liquid. I believe that this manuscript can be considered for publication, but requires revision.
1. This study proposes a new working fluids, refrigerant/IL+ carbon nanomaterials (CNMs). Compared with traditional working fluids, what are the advantages of this new type of working fluids?
2. The authors mentioned in the “introduction” part that “Investigations related to the application of ammonia/ionic liquid working fluids in absorption refrigeration systems are limited in the open literature. Moreover, studies on the absorption refrigeration systems working with ammonia/ionic liquid+nanomaterials working fluids are not reported in the literature.” And they also emphasized in the “Results and discussions” part that “No data for the thermal properties of the [Bmim]BF4 +CNMs mixed with NH3 solutions have been reported in the literature.” Since ionic liquids are considered a novel type of green working fluid that can be used in various fields, why is there so little research on it? Is it because the study of this material experimentally difficult? Or because the application prospects of this working liquid are not so good?
3. The authors did not introduce the experimental methods in this manuscript. What theories or models are used in this study? Please introduce briefly in the revised manuscript.
Author Response
Response to comments of Reviewer 1
Thank to Reviewer for comments and suggestions. We have considered your suggestions and we revised the manuscript according to your comments.
This manuscript analyzed and discussed the thermo-physical properties of the ammonia with graphene and single-wall carbon nanotubes respectively dispersed into [Bmim]BF4 ionic liquid, systematically. The correlations of the thermal conductivity, viscosity, specific heat, and density of the NH3/IL+ CNMs solutions are proposed. This study may provide a good reference for experimental researches in the ionic liquid. I believe that this manuscript can be considered for publication, but requires revision.
Comment 1: This study proposes new working fluids, refrigerant/IL+ carbon nanomaterials (CNMs). Compared with traditional working fluids, what are the advantages of this new type of working fluids?
Response 1: Usually, ILs have high boiling points, strong affinities with refrigerants and favorable thermal and chemical stabilities. They provide an alternative way to prevent the risks of corrosion and crystallization of H2O/LiBr pair and weaknesses of low efficiency and complexity of systems with NH3/H2O pair.
Comment 2: The authors mentioned in the “introduction” part that “Investigations related to the application of ammonia/ionic liquid working fluids in absorption refrigeration systems are limited in the open literature. Moreover, studies on the absorption refrigeration systems working with ammonia/ionic liquid+nanomaterials working fluids are not reported in the literature.” And they also emphasized in the “Results and discussions” part that “No data for the thermal properties of the [Bmim]BF4 +CNMs mixed with NH3 solutions have been reported in the literature.” Since ionic liquids are considered a novel type of green working fluid that can be used in various fields, why is there so little research on it? Is it because the study of this material experimentally difficult? Or because the application prospects of this working liquid are not so good?
Response 2: In the last years, the NH3/ILs solutions have been proposed as working fluids in absorption cycles to remove rectification and distillation sections, and to improve cycles’ thermal efficiency. Applications of these fluids used in cycles for heating and cooling introduced in previous studies [1, 2] also confirm their promising potentials. Moreover, recent studies showed that by adding nanomaterials in the ionic liquids, their properties are enhanced and thus thermal efficiency of the absorption refrigeration systems will be enhanced as well.
[1] M. Wang, T.M. Becker, B.A. Schouten, T.J. Vlugt, C.A. Infante Ferreira, Ammonia/ ionic liquid based double-effect vapor absorption refrigeration cycles driven by waste heat for cooling in fishing vessels, Energy Convers. Manage. 174 (2018) 824–843.
[2] M. Wang, C.A. Infante Ferreira, Absorption heat pump cycle with NH3 – ionic liquid working pairs, Appl. Energy 204 (July) (2017) 819–830.
Comment 3: The authors did not introduce the experimental methods in this manuscript. What theories or models are used in this study? Please introduce briefly in the revised manuscript.
Response 3: Thermo-physical properties of the studied NH3/IL+CMNs mixtures have been computed on the basis Eq. (1). This equation provides a general form of ideal solution properties, which is based on a weighted average of properties from both components and is used for estimating properties in several papers [26, 27].
We thank you for your support.
Sincerely yours,
Authors

Reviewer 2 Report
The authors introduce thermo-physical properties of the NH3/IL+ carbon nanomaterials solutions. I have carefully read the manuscript, and I found that this is an interesting work with promising results, and the structure and logic are good. I am happy to recommend this work to be published in this journal after some revisions since there are still many unclear points in the current manuscript. My comments listed below may help the authors further improve their work:
- The current version failed to tell the readers what are the innovations of the work. Please strengthen.
- In the introduction and background parts, the authors should concisely summarize the progress on the development of the research field.
- How to evaluate the specific heat?
- This work provides lots of data, but a clear story with good logic should be given. Some figures can be combined to state the sub-stories.
- Some further outlooks are needed at the end of the manuscript.
- Please use only 1 paragraph as your conclusion.
- Many expression in this work is not scientific, the English should be further polished.
Author Response
Response to comments of Reviewer 2
Thank to the Reviewer for comments and suggestions. We have considered your suggestions and we revised the manuscript according to your comments.
The authors introduce thermo-physical properties of the NH3/IL+ carbon nanomaterials solutions. I have carefully read the manuscript, and I found that this is an interesting work with promising results, and the structure and logic are good. I am happy to recommend this work to be published in this journal after some revisions since there are still many unclear points in the current manuscript. My comments listed below may help the authors further improve their work:
Comment 1: The current version failed to tell the readers what are the innovations of the work. Please strengthen.
Response 1: The paper proposes the use of new working fluids in absorption refrigeration systems in order to improve their thermal efficiency. Since there no are studies reported in the literature on the absorption refrigeration systems working with ammonia/ionic liquid+nanomaterials solutions, thermo-physical properties of the working fluids being the main data in the evaluation of the performance of those systems, this paper presents a set of data regarding the thermo-physical properties of NH3/IL+CNMs.
These arguments are including in the manuscript.
Comment 2: In the introduction and background parts, the authors should concisely summarize the progress on the development of the research field.
Response 2: The progress on the development of the research field concretized by the coefficient of performance of the absorption refrigeration systems using various NH3/ionic liquids is summarized in Table 1. Regarding the use of NH3/IL+NMs there no are data available in the literature.
Comment 3: How to evaluate the specific heat?
Response 3: The specific heat of NH3/IL+CNMs is computed using the Eq. (1). This equation was used also in Ref. [26] for computing the specific heats of several NH3/IL mixtures and has been verified with data of H2O/[mmim][DMP] from Ref. [3].
[3] Dong L, Zheng D, Nie N, Li Y. Performance prediction of absorption refrigeration cycle based on the measurements of vapor pressure and heat capacity of H2O +[DMIM]DMP system. Appl Energy 98 (2012)326–332.
Comment 4: This work provides lots of data, but a clear story with good logic should be given. Some figures can be combined to state the sub-stories.
Response 4: From my point of view, an overlapping of some figures would lead to difficulty to ‘read” of results. The results are plotted for three working fluids, each working fluid with two concentrations of CNMs, eight NH3 fractions, and six temperatures.
Comment 5: Some further outlooks are needed at the end of the manuscript.
Response 5: The results of this study may contribute to the consolidation of the property database of NH3/IL+NMs for absorption refrigeration applications. Further investigations concerning the thermo-physical characteristics of ammonia with other types of ionic liquids are needed. Also, for practical implementation of NH3/ILs+CNMs in the absorption refrigeration systems the experimental studies to support the reported theoretical results are needed.
Comment 6: Please use only 1 paragraph as your conclusion.
Response 6: The “Conclusions” part was rearranged in the revised manuscript.
Comment 7: Many expression in this work is not scientific, the English should be further polished.
Response 7: We have checked the language and we performed several improvements. Thank you!
We thank you for your support.
Sincerely yours,
Authors

Round 2
Reviewer 2 Report
The current version is OK for publication.